# Novel kind of decagonal ordering in Al$_{74}$Cr$_{15}$Fe$_{11}$

Haikun Ma[1], Zhanbing He[1✉], Hua Li[1], Tiantian Zhang [1], Shuang Zhang[2], Chuang Dong[2,3] & Walter Steurer[4]

A high-angle annular dark field scanning transmission electron microscopy study of the intermetallic compound Al$_{74}$Cr$_{15}$Fe$_{11}$ reveals a quasiperiodic structure significantly differing from the ones known so far. In contrast to the common quasi-unit-cells based on Gummelt decagons, the present structure is related to a covering formed by Lück decagons, which can also be described by a Hexagon-Bow-Tie tiling.

[1] State Key Laboratory for Advanced Metals and Materials, University of Science and Technology Beijing, Beijing 100083, China. [2] School of Materials Science and Engineering, Dalian Jiaotong University, Dalian 116028, China. [3] Key Laboratory for Materials Modification by Laser, Ion and Electron Beams (Dalian University of Technology), Ministry of Education, Dalian 116024, China. [4] Department of Materials, ETH Zurich, 8093 Zurich, Switzerland. ✉email: hezhanbing@ustb.edu.cn

Quasiperiodic structures can be described geometrically either as decorated tilings or as coverings (for a general review, see ref. [1], for instance). Although quasiperiodic tilings are based on two or more unit tiles, coverings can cover the plane (space) by partially overlapping copies of a single structural repeat unit (quasi-unit-cell[2–6]). Such a quasi-unit-cell, when decorated with atoms (atomic cluster), is the counterpart to a unit cell of a periodic structure. It allows a more physical description of a quasicrystal structure than the structurally fully equivalent tiling-based models do[6–8]. In the case of decagonal quasicrystals (DQCs), the so-far most frequently used quasi-unit-cell is based on the Gummelt decagon[9–12], if the quasi-unit cell approach is employed at all.

In this work, we present a quasiperiodic structure that can be only described by a covering based on the Lück decagon, a kind of an experimentally identified quasi-unit-cell for a decagonal phase. This fundamental decagonal unit, with a diameter of ~2 nm in our case, consists of four subunits: three flattened hexagons and one bowtie ($D_{3H+1BT}$, for short).

## Results

**Characteristic SAED patterns of decagonal $Al_{74}Cr_{15}Fe_{11}$.** It was previously shown that samples with the nominal composition $Al_{72}Cr_{16}Fe_{12}$ contain a DQC[13]. As its crystal structure has been not determined so far, we decided to carry out a detailed study of this DQC. In order to get samples of highest quality, we did a series of annealing experiments (see the "Methods" section for details). The best single-crystal of the DQC was identified by selected-area electron diffraction (SAED) and its composition determined to $Al_{74}Cr_{15}Fe_{11}$ by energy dispersive X-ray spectrometry (EDS) in a transmission electron microscope. Figure 1 shows the SAED patterns along the tenfold axis (Fig. 1a) and along the two typical twofold zone axes, D and P, perpendicular to it (Fig. 1b, c). Some characteristic features of the diffraction pattern of a DQC, such as its scaling symmetry, are visualized by pentagons of varying size in Fig. 1a. It is noteworthy that the spots marked by yellow circles in the SAED pattern are much weaker than the other spots of the largest pentagon, analogously to that of decagonal Al-Cr-Fe-Si[5], but quite different from other typical Al-based DQCs such as Al-Ni-Co[14,15], where only strong diffraction spots are found in the corresponding positions. The translation period of the $Al_{74}Cr_{15}Fe_{11}$ DQC, determined from the two twofold SAED patterns, is ≈1.23 nm, comparable to that of decagonal Al-Mn-Pd[16]. Consequently, the structure has a translation period of six quasiperiodic atomic layers stacked periodically along the tenfold axis. Every other reciprocal lattice layer is systematically extinct in Fig. 1c indicating the existence of a $c$-glide plane and a $10_5$ screw axis in the five-dimensional (5D) embedding space. Thus, the 5D symmetry group of this DQC should be $P10_5/mmc$, such as for decagonal Al-Mn-Pd[16].

**Covering based on Lück decagons for decagonal $Al_{74}Cr_{15}Fe_{11}$.** The high-angle annular dark field-scanning transmission electron microscopy (HAADF-STEM) image, which corresponds to a projection of the structure along the tenfold direction, is depicted in Fig. 2a, b. Connecting related dots of the image yields a Hexagon-Bow-Tie (HBT) tiling (Fig. 2a), a tiling that was first studied by Lück[17] and by Lück and Lu[18]. This tiling can be described by a covering as well (Fig. 2b). The covering cluster is a decagon of approximately 2 nm diameter, partitioned by three flattened hexagon (H) tiles and one bowtie (BT) tile, called Lück decagon by us ($D_{3H+1BT}$, for short). An example for a covering created by copies of the Lück decagon is shown in the tilings encyclopedia[19]. In that example, originally created by Andritz, the underlying HBT tiling is a substitutional tiling, which scales by even powers of $\tau$. In contrast to the Gummelt covering, connecting the decagon centers results in a somehow randomized Tübingen-triangle-tiling (TTT)[20] and not in a pentagon-Penrose tiling (P1-PT) (see Fig. 2d). Figure 2d is an idealized quasiperiodic covering based on $D_{3H+1BT}$ decagons without gaps, derived from Fig. 2b. The gaps of purple BT and star (S) tiles patched in the quasi-unit-cell matrix in Fig. 2b are eliminated through the action of phason flipping (will discuss later).

The $D_{3H+1BT}$ decagons are superposed onto the HAADF-STEM image in Fig. 2a and are filled in light-blue for clear display in Fig. 2b. It is noteworthy that the vertices of the $D_{3H+1BT}$ decagons are themselves decorated with smaller decagons (see also the projected structural model in Fig. 3b, where the centers of the smallest D clusters (~0.47 nm in diameter) are assigned to heavier elements such as Cr, Fe because of their higher intensities in the HAADF-STEM image[21]), but not all are exactly the same. For the vertices of $D_{3H+1BT}$ (or H) to jointly form the thin waist of nearby BT tile, their atomic decorations are not the same as that of the other vertices decorated with the smaller decagons. The covering by $D_{3H+1BT}$ is not perfect; some gaps are left marked as purple BT and S tiles. The percentage of area covered by gaps is 6.1% of the whole filled area in Fig. 2b and can be totally eliminated through the action of phason flipping, e.g., those in Fig. 2c and Supplementary Fig. 1.

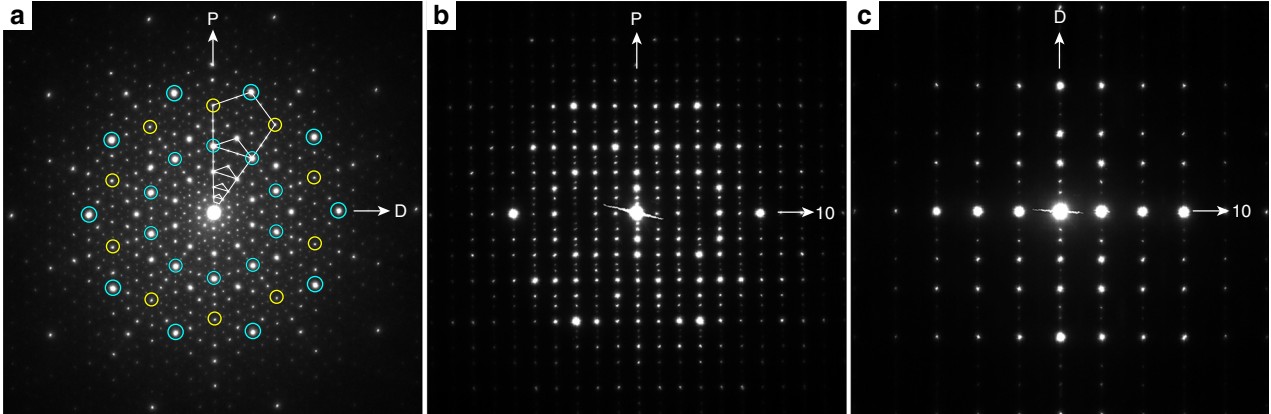

**Fig. 1 Characteristic SAED patterns of $Al_{74}Cr_{15}Fe_{11}$. a** Taken along the tenfold axis and **b**, **c**, along two typical twofold zone axes, D and P, normal to that in **a**. The diffraction pattern in **a** shows an evident tenfold symmetry, e.g., those marked by blue circles. Some features are marked in **a** demonstrating the scaling properties of the diffraction pattern by powers of $\tau = 1.618$, as visualized by pentagons of varying size. Every other reciprocal lattice layer is systematically extinct in **c**, indicating the existence of a $c$-glide plane and, intrinsically, a $10_5$ screw axis.

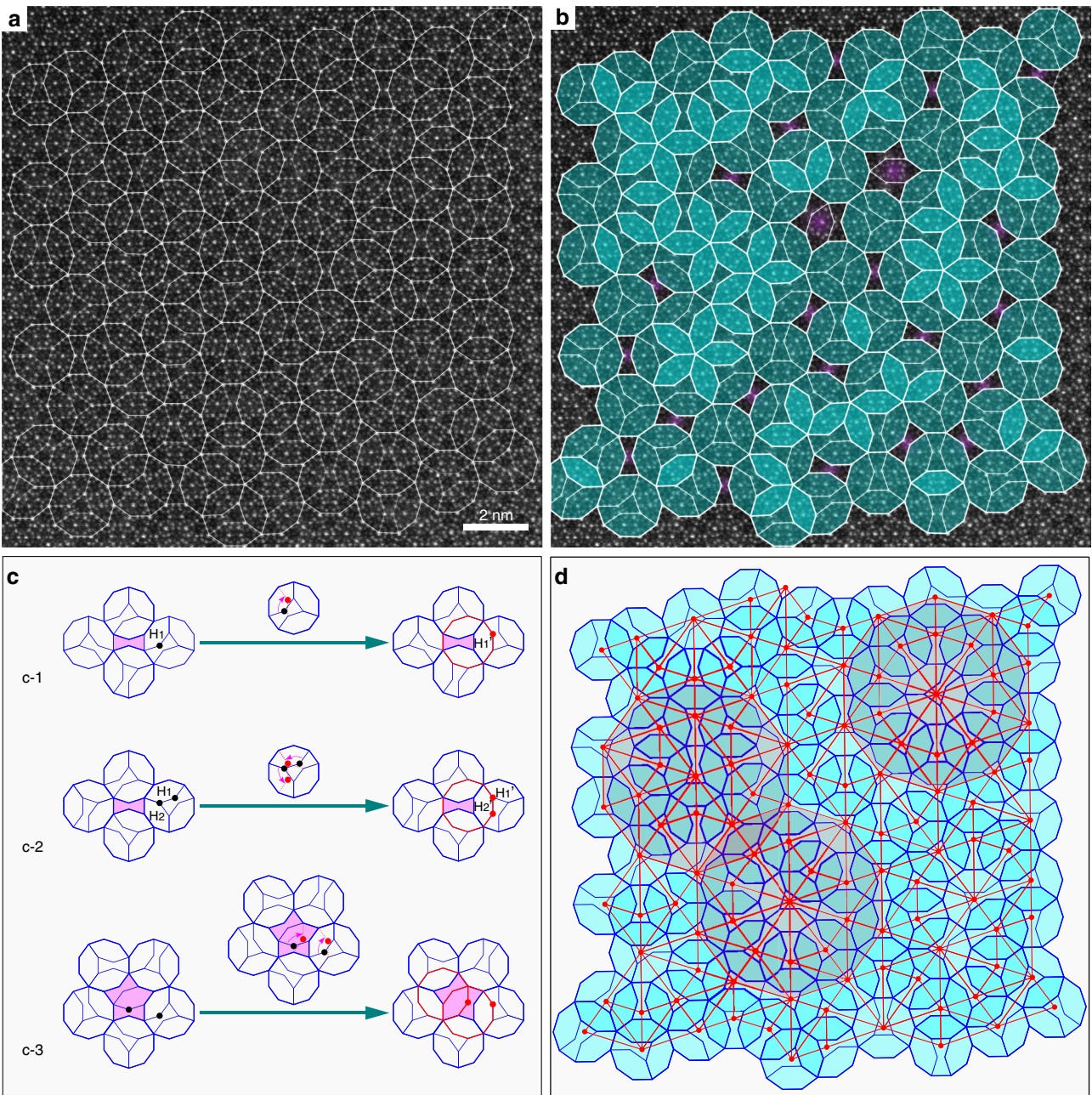

**Fig. 2 Covering based on the Lück decagon. a** HAADF-STEM image along the tenfold axis of decagonal $Al_{74}Cr_{15}Fe_{11}$ with a Lück-decagon covering superposed. **b** The Lück decagons are shaded blue to clearly show some defective areas (6.1% of the total area) marked in purple. **c** The action of phason flipping to eliminate the defects (purple area) visible in **b**. One or two more Lück decagons are generated in the right column with the disappearance of the purple defects in the left column, after the change of few flat hexagonal (H) tiles. **d** If the centers of the Lück decagons (red dots) are connected by lines a triangle tiling is created, which resembles the Tübingen-triangle tiling (TTT). For the TTT typical decagonal areas are shaded in transparent red. In the ideal TTT the straight lines would not be broken.

The purple BT gap in the left column of Fig. 2c-1 is eliminated by being included in a quasi-unit-cell of $D_{3H+1BT}$ decagon (outlined red) in the right column, after the change of $H_1$ tile in the left column to the $H_1'$ tile in the right column. In fact, the change from $H_1$ to $H_1'$ tile is simply realized through the phason flipping by changing only one vertex (namely, from black spot to the red spot, as marked by a red arrow in the middle), which was observed experimentally through in situ high resolution transmission electron microscopy (TEM) observations[22,23]. Consequently, the defect of purple BT in the left column is mended and one more quasi-unit-cell is accordingly added, as shown by the red $D_{3H+1BT}$ in the right column. Sometimes, the phason flipping

is somewhat more complex than that in Fig. 2c-1. For example, to eliminate defect of purple BT in Fig. 2c-2 in the left column and to also maintain the quasiperiodic repeating of the nearby quasi-unit-cell of $D_{3H+1BT}$ tiles without gaps, the change of two vertexes of tiles is needed, and resulting in the change of $H_1$ and $H_2$ tile in the left column to the $H_1'$ and $H_2'$ tile in the right column. Occasionally, the change of three atomic positions is needed (see in Supplementary Fig. 1a). For the defect of S tiles in Fig. 2c-3 in the left column, two atomic positions are changed and create two more $D_{3H+1BT}$ tiles (also see another example in Supplementary Fig. 1b), rather than one more $D_{3H+1BT}$ tile for eliminating BT-type defects mentioned above.

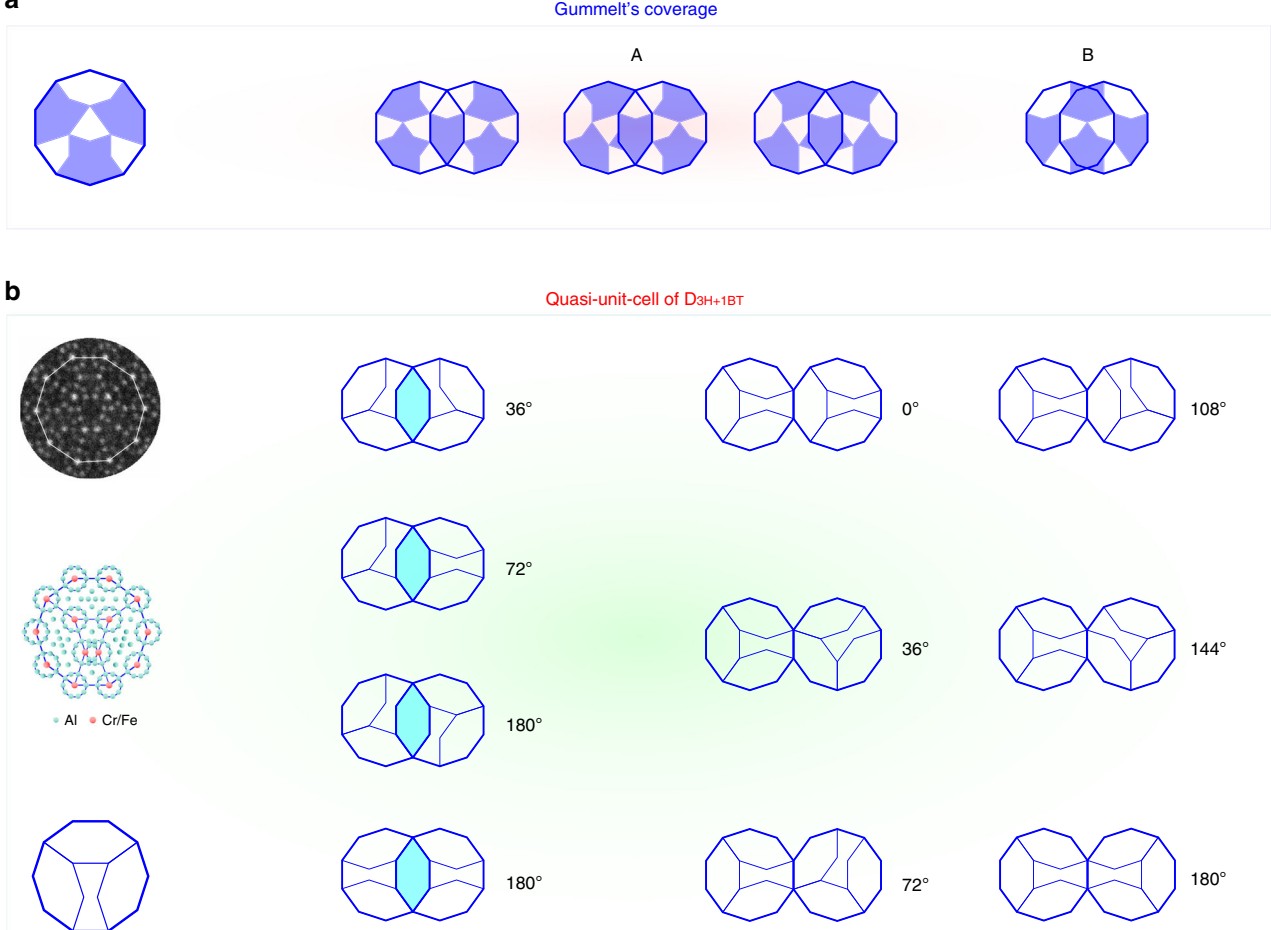

**Fig. 3 Comparison of overlaps of Gummelt decagons and of the experimentally observed Lück decagons. a** Allowed arrangements of Gummelt decagons. Middle and right: A- and B-type overlaps with strict matching rules (see ref. [9] for details). **b** Quasi-unit-cell of $D_{3H+1BT}$ in this paper. Left column: experimental image, projected structural model (Red atoms: Cr/Fe; blue atoms: Al) and schematic diagram of the quasi-unit-cell $D_{3H+1BT}$ ($\approx$ 2.0 nm diameter). Middle and right: typical connections of Lück decagons by overlapping H tiles or by sharing an edge. They lead to two different distances of the decagon centers: $S = 1.18$ nm, and $L = \tau S = 1.91$ nm. Their connections are relatively simple without strict matching rules. The observed orientations of adjacent decagons are shown aside.

## Discussion

**Comparison of Gummelt decagons and Lück decagons**. The different allowed arrangements of Gummelt decagons in a strictly quasiperiodic covering (Penrose tiling) are compared to the experimentally observed ones of the Lück decagons in Fig. 3a. The Lück decagon ($D_{3H+1BT}$) in Fig. 3b is generated by linking the centers of the ten smaller rings (with a diameter of ~ 0.47 nm) with tenfold symmetry. Four more of these small rings located inside the $\approx$ 2 nm decagon, form the three shuttle-like H tiles and one BT tile with an edge length of $\approx$ 0.62 nm. The $D_{3H+1BT}$ has just mirror symmetry, similar to the Gummelt's decagon. How-ever, although the reflection plane runs through corners in the case of the Gummelt decagon, it is perpendicular to decagon edges in the case of the Lück decagon.

In contrast, the $D_{3H+1BT}$ are linked to their neighbors by either overlapping H tiles or sharing one edge (Fig. 3b). The distance between the centers of adjacent $D_{3H+1BT}$ decagons with over-lapping H tiles amounts to $S = 1.18$ nm and to $L = \tau S = 1.91$ nm when the $D_{3H+1BT}$ decagons are sharing edges. The angles between the two adjacent $D_{3H+1BT}$ decagons with overlapping H tiles are $\theta = n \times 36°$ ($n = 1, 2, 5$), and $\theta = n \times 36°$ ($n = 0, 1, 2, 3, 4, 5$) when the $D_{3H+1BT}$ decagons are sharing edges. For the linkage of more $D_{3H+1BT}$ decagons, e.g., three, four, five and more

$D_{3H+1BT}$ tiles see Supplementary Fig. 2. The simple connection rules in Fig. 3b allow covering the whole plane without gaps, implying the $D_{3H+1BT}$ can act as a repeating quasi-unit-cell generating a quasiperiodic long-distance order[19]. Therefore, the quasi-unit-cell of $D_{3H+1BT}$ is totally different from Gummelt's decagon. First, there are no Gummelt's B-type overlaps of the decagons in decagonal Al-Cr-Fe. Second, the A-type overlaps are not polar in our case in contrast to those of Gummelt decagons, where strictly matching rules have to be obeyed. Lastly, the connections of nearby $D_{3H+1BT}$ tiles through sharing one edge are largely found in our case, which is not allowed in the case of Gummelt decagons. Therefore, Gummelt decagons cannot be applied to define the structure of this quasicrystal.

**Creation of a tiling by linking the centers of Lück decagons**. We now analyze the local features and the long-distance quasi-periodic ordering of the quasi-unit-cell of $D_{3H+1BT}$ tiles in Fig. 2d by linking their centers with solid green lines of equal length (Fig. 4a), to check how close the resulting tiling is to a P1-PT. Although there are regular pentagons (P), the large decagons, fat hexagons ($H_F$), banana-like tiles, and concave decagons ($D_C$) are no parts of a P1-PT. Irregular polygons were mostly found in

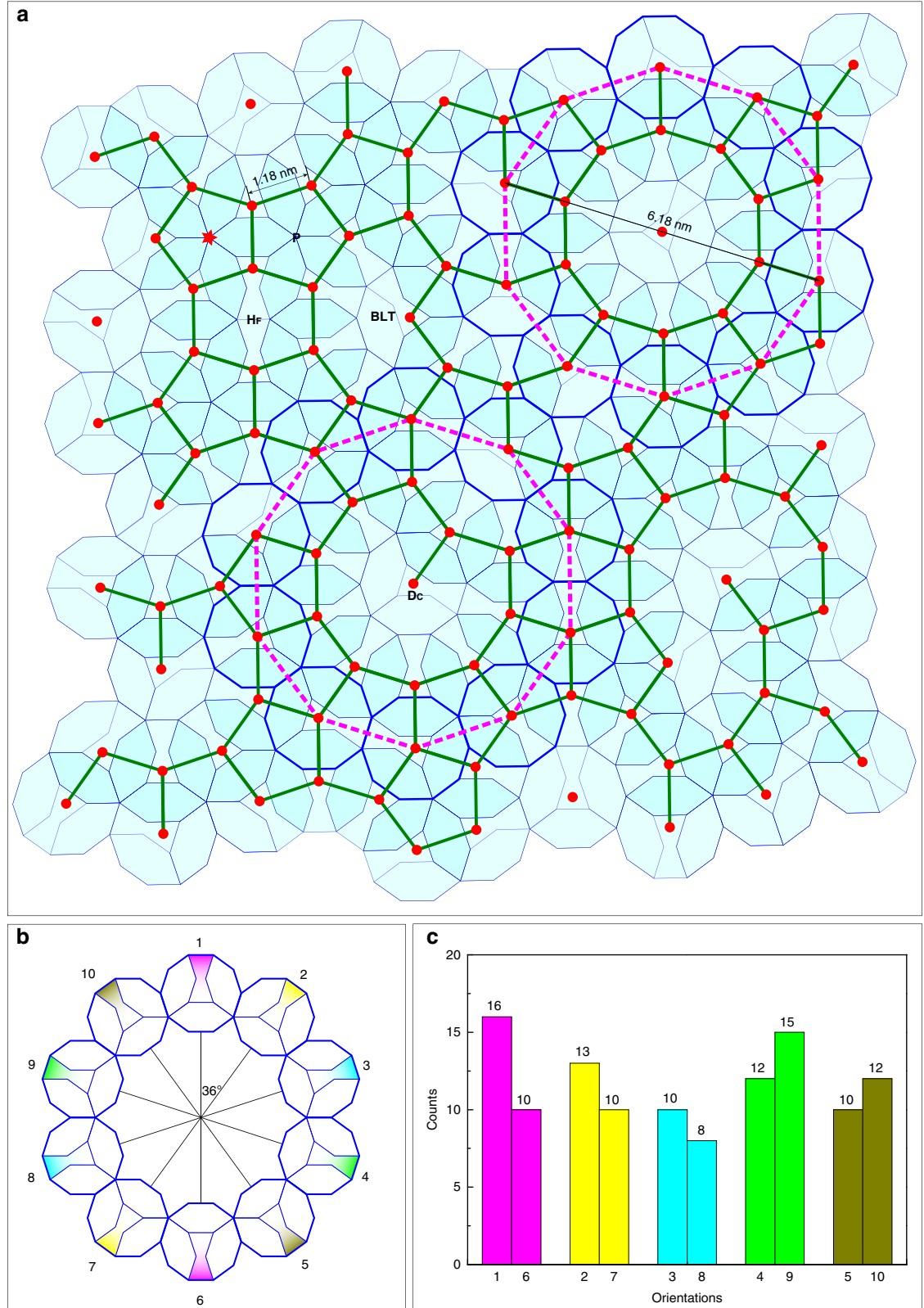

**Fig. 4 Creation of a tiling by linking the centers of overlapping Lück decagons.** In case of a Gummelt-decagon covering a P1-PT would result, what is not the case here. **a** The set of structural tiles contains regular pentagons (P), large decagons, fat hexagons ($H_F$), banana-like tiles (BLT), and concave decagons ($D_C$). The large purple dotted decagons (6.18 nm in diameter) correspond to the large shaded decagons in Fig. 2d. **b** The ten orientations of the Lück decagons observed in **a**. **c** Corresponding histogram of decagon counts along each orientation. The average count of decagons along each direction is 11.1 ± 2.1.

imperfect DQCs[24–27], implying the quasiperiodic ordering in Fig. 4a is not perfect. The arrangement of ten $D_{3H+1BT}$ decagons (highlighted with thick blue edges) at the vertices of the larger decagon (dotted purple lines) with a diameter of 6.18 nm is generated. These edge-sharing $D_{3H+1BT}$ decagons resemble distribution of ≈ 2.0 nm decagons in decagonal $Al_{59}Cr_{21}Fe_{10}Si_{10}$[28]. Furthermore, we note that all the 116 decagons in Fig. 4a are distributed along 10 directions differing by 36°. Figure 4b schematically shows the directions of decagons, where the BT tile in each decagon is filled by colors to guide the eye. Among them, every two of 180° oriented decagons are in a pair and colored in the same. We count the decagons along each direction and summarize in the histogram in Fig. 4c, where the maximum, minimum, and averaged number of decagons is 16, 8, and 11.1 ± 2.1, respectively. The difference of the counts of decagons along each direction also implies the imperfect quasiperiodic tiling.

A detailed theoretical discussion of the diversities and communalities of these two basic decagonal clusters, the Gummelt and the Lück decagon, and their coverings is beyond the scope of the present paper. This will be the topic of a forthcoming paper by Steurer. Finally, what are the driving forces for the formation of this kind of quasicrystal? It is generally accepted nowadays that quasiperiodic structures and their rational approximants result from packing energetically favorable structural subunits called clusters. Due to the non-crystallic symmetry of these clusters they have to "overlap," to avoid gaps between them. "Overlapping" means that each cluster already contains a part of the adjacent cluster in itself. There are some other factors contributing to the formation of quasiperiodic structures that have been discussed in greater detail elsewhere[29]. This not only works for Gummelt decagons but for Lück decagons as well.

In summary, the finding of this Lück-decagon-based quasi-unit-cell shows that quasiperiodic order in real DQCs is possible based on a variety of fundamental structural subunits (quasi-unit cells). Both, the Gummelt as well as the Lück covering have in common that their actual three-dimensional structural subunits are arranged along quasi-lattice planes with traces forming Fibonacci pentagrids. These planes may be important for the evolution of quasiperiodic long-range order. The main difference between the two types of coverings lies in the underlying tilings, P1-PT in the case of Gummelt decagons and TTT in the case of Lück decagons.

## Methods

**Preparation of quasicrystals**. Approximately 1 kg of the master alloy with a nominal composition of $Al_{72}Cr_{16}Fe_{12}$ consisting of high-purity elements was first molten in a ZG-001 induction furnace (Liaoning Jinzhou Electric Furnace Co., Ltd) at a temperature of 1500 °C under vacuum. Then the molten alloy was poured into a graphite crucible in the furnace to form an ingot. Several small pieces from the cast ingot were sealed in evacuated quartz tubes for heat treatments in a SX-G04133 electric box furnace (Tianjin Zhonghuan Furnace Corp.). Finally, we found that two heating processes are necessary for producing a DQC of highest quality: first, annealing at 1025 °C for 7 days and then cooling in the furnace after switching off the power and, second, annealing at 1000 °C for 7 days, followed by quenching in water.

**Characterizations**. Powder samples were adapted for TEM observations. We first crushed small blocks from the ingot into powders. Then, alcohol was added for preparing a suspension for a 3 min ultrasonic treatment. Finally, for the TEM observations a drop of the suspension was dripped onto a 3 mm copper grid covered by hollow carbon film. A FEI Tecnai F30 transmission electron microscope equipped with an EDS was first used to check the phases and the composition. A JEM-ARM200F transmission electron microscope equipped with a Cs-probe corrector and Cs-image corrector was used subsequently to obtain HAADF-STEM images at an atomic resolution. The inner and outer acceptance semi-angles were 90 and 370 mrad, respectively.

## Data availability

The data that support the findings of this study are available from the corresponding author upon reasonable request.

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

## Acknowledgements

This work was supported by the National Natural Science Foundation of China (51871015 and 51471024). We thank Mr. Xinan Yang of the Beijing National Laboratory

for Condensed Matter Physics, Institute of Physics, Chinese Academy of Sciences, for assistance in recording the HAADF-STEM images.

## Author contributions

Z.H. conceived the research. H.M. performed the experiments. Z.H., H.M., and W.S. wrote the manuscript. H.M., Z.H., H.L., T.Z., S.Z., C.D., and W.S. analyzed the data, discussed the results, and drew the conclusions.

## Competing interests

The authors declare no competing interests.
