## [Peer Review File · Nature Communications]

REVIEWER COMMENTS

Reviewer #1 (Remarks to the Author):

This manuscript reports a novel kind of decagonal ordering in Al₇₄Cr₁₅Fe₁₁ alloy system. The experimental results are worthy of being published.

Please find comments and suggestions on the improvement of the manuscript,

1. Tiling is formed by selected the spots in the HAADF image as shown in Figure 2(b). The flat hexagonal tiles in the tilting were not identical to the each other in some areas, which means the tilting is not the unique figure for the tiling description of the HAADF image. The authors need to explain the way to they made the tiling, and how to avoid the other possible choice.
2. Atom positions of the local cluster should be given to support the tiling model. The rule on the local cluster to form the tilting should be discussed.

Reviewer #2 (Remarks to the Author):

This is an experimental study about an Al-Cr-Fe decagonal quasicrystal where the authors have shown, through HAADF-STEM analysis, the assembly of a new type of quasi-unit-cell occurring in this quasicrystalline phase. In this paper the authors define the new quasiperiodic structure as a covering, based on Andritz decagon rather than the Gummelt decagon, usually used to define decagonal quasi-unit-cells. This is, in principle, an important result because a new unit-cell is proposed and experimentally shown. The reviewer finds the results technically sound and supported by the data provided and believes that it will be of interest for researchers studying quasicrystal crystallography. However, there are certain aspects of the manuscript that, in the reviewer's opinion, must be addressed prior to the possibility of acceptance, as follows:

Major concerns

- 1) Authors have shown that the combination of 3 flattened hexagons and one bow-tie can, with the aid of phason flipping, fill the 10-fold symmetry plane. I believe that authors should show that the previously known quasi-unit-cells (e.g., through Gummelt decagon) cannot be applied to define the structure of this quasicrystal. Otherwise, one could argue that the authors only found a new way of representing this structure, and not in fact, a different structure (such as in the case of crystals: we can always find a primitive unit cell to describe a centered unit cell, however, it does not makes it two different structures).
- 2) Authors should stress more strongly why this is an important finding and what advances can it bring to the scientific community.

Minor concerns

- 1) Methods section is extremely short and lacks in detail. For instance, why was the Al₇₂Cr₁₆Fe₁₂ composition chosen? What was the induction furnace model and manufacturer? What temperature was used for the melting process? Did the authors check the final chemical composition of the sample? Why two annealing temperatures? Powder samples were used for TEM analysis, but what was the sample preparation procedure? Authors say that EDS was used to check the chemical composition during TEM analysis. What was the chemical composition of the quasicrystal (not the nominal, but real

composition)?

2) In figure 1c authors claimed the existence of a c-glide due to extinctions and the authors defined the 5D symmetry as P105/mmc. Authors should detail the diffraction features that led to the conclusion of a presence of the screw-axis as well.

REVIEWER COMMENTS

Reviewer #1 (Remarks to the Author):

This manuscript reports a novel kind of decagonal ordering in $\text{Al}_{74}\text{Cr}_{15}\text{Fe}_{11}$ alloy system. The experimental results are worthy of being published.

Please find comments and suggestions on the improvement of the manuscript,

1. Tiling is formed by selected the spots in the HAADF image as shown in Figure 2(b). The flat hexagonal tiles in the tilting were not identical to the each other in some areas, which mean the tilting is not the unique figure for the tiling description of the HAADF image. The authors need to explain the way to they made the tiling, and how to avoid the other possible choice.

Response: Thank you for your valuable and thoughtful comments. It is true that the a few flat hexagonal (H) tiles in the tilting were not identical to the most H tile. The way to make the tiling is by connecting the centers of smallest decagons (with a diameter of around 0.47 nm), which will produce general structural blocks found in DQCs, such as H, BT, B, and D tiles, as we described in the paper. However, structural disorder could happen to a few H tiles, which makes a few H tiles are not identical to the most others in some areas.

2. Atom positions of the local cluster should be given to support the tiling model. The rule on the local cluster to form the tilting should be discussed.

Response: We agree with the reviewer that it should be quite good if atom positions of the local cluster could be given. Unfortunately, we cannot determine the atom positions for the moment because of lacking the structural information of Al-Cr-Fe DQC. The crystal structures of Al-Cr-Fe DQC could be determined by X-ray single crystal diffraction, which might be done in our future works.

The rule on the local cluster to form the tilting as follows. The D_{3H+1BT} are linked to their neighbors by either overlapping H tiles or sharing one edge. The distance between the centers of adjacent D_{3H+1BT} decagons with overlapping H tiles amounts to $S = 1.18$ nm, and to $L = \tau S = 1.91$ nm when the D_{3H+1BT} decagons are sharing edges. The angles between the two adjacent D_{3H+1BT} decagons with overlapping H tiles are $\theta = n \times 36^\circ$ ($n = 1, 2, 5$) for overlapping H tiles, and $\theta = n \times 36^\circ$ ($n = 0, 1, 2, 3, 4, 5$) or for sharing one edge. We added some more contexts about this point on P7 in the revised version.

Reviewer #2 (Remarks to the Author):

This is an experimental study about an Al-Cr-Fe decagonal quasicrystal where the authors have shown, through HAADF-STEM analysis, the assembly of a new type of quasi-unit-cell occurring in this quasicrystalline phase. In this paper the authors define the new quasiperiodic structure as a covering, based on Andritz decagon rather than the Gummelt decagon, usually used to define decagonal quasi-unit-cells. This is, in principle, an important result because a new unit-cell is proposed and experimentally shown. The reviewer finds the results technically sound and supported by the data provided and believes that it will be of interest for researchers studying quasicrystal crystallography. However, there are certain aspects of the manuscript that, in the reviewer's opinion, must be addressed prior to the possibility of acceptance, as follows:

Major concerns

1) Authors have shown that the combination of 3 flattened hexagons and one bow-tie can, with the aid of phason flipping, fill the 10-fold symmetry plane. I believe that authors should show that the previously known quasi-unit-cells (e.g., through Gummelt decagon) cannot be applied to define the structure of this quasicrystal. Otherwise, one could argue that the authors only found a new way of representing this structure, and not in fact, a different structure (such as in the case of crystals: we can always find a primitive unit cell to describe a centered unit cell, however, it does not makes it two different structures).

Response: Thank you very much for your valuable and thoughtful comments which help us improve the quality of paper. We added some more text to explain why previously known quasi-unit-cell of Gummelt decagon cannot be applied to the Al-Cr-Fe DQCs. Note that we changed the term 'Andritz decagon' to 'Lück decagon' because we found that Lück was the very first discussing the hexagon-bow-tie tiling.

The 2 nm D clusters in Al-Cr-Fe DQCs are different from the one in the previously reported DQCs such as in Al-Co-Ni systems, in which Gummelt's decagons were usually used to describe their tiling structures. [For example, Steinhardt, P. J. et al. Experimental verification of the quasi-unit-cell model of quasicrystal structure. *Nature* **396**, 55-57 (1998); Abe, E. Electron microscopy of quasicrystals—where are the atoms. *Chem. Soc. Rev.* **41**, 6787 (2012).]. In our case, the decagons are partitioned by three flattened hexagon tiles and one bowtie tile

(D_{3H+1BT} , for short), and the linkage of the centers of the ten smallest D clusters with a tenfold symmetry will generate the profile of the D_{3H+1BT} . It is totally different from Gummelt's decagon, resulting in the invalidation of Gummelt's coverage. Firstly, there is no B-type coverage of Gummelt's decagons in the Al-Cr-Fe decagonal quasicrystal. Secondly, A-type coverage in our case is more flexible, also different from the A-type coverage of Gummelt's decagons, where strictly matching rules have to be obeyed. Lastly, the connections of nearby D_{3H+1BT} tiles through sharing one edge are largely found in our case, which is not allowed by the rules of Gummelt's coverage. Therefore, Gummelt's decagon cannot be applied to define the structure of this quasicrystal. We added some more discussion about this point on P7 in the revised version.

2) Authors should stress more strongly why this is an important finding and what advances can it bring to the scientific community.

Response: Compare to the two unit-tiles of Penrose models, the quasi-unit-cell model with a single structural repeat-unit is simpler because we only need to consider the atomic rearrangements within a single quasi-unit cell in real. The quasi-unit cell of Gummelt's decagon was proposed from a geometric point of view and was applied successfully in some decagonal quasicrystals, for example in typical Al-Co-Ni system. However, strict matching rules of Gummelt decagons limited its wide applications. For experimental quasicrystals, the crystal structures of decagons and the corresponding tilings are diverse, and lots of them can not be well described by Gummelt's decagon. Therefore, new quasi-unit-cells are highly desired to describe more general cases of quasicrystals, especially for most observed imperfect quasicrystals. Experimentally, we found a quasi-unit-cell of D_{3H+1BT} , the type of Andritz decagon, in Al-Cr-Fe system, which is totally different from Gummelt decagons, but with advantages of combining tiling and covering. Consequently, the quasi-unit-cell of D_{3H+1BT} has greater flexibility and without strict connecting rules, which might produce rich long-distance quasiperiodic orderings, corresponding to most experimental observations of DQCs. In addition, the quasi-unit-cell of D_{3H+1BT} in Al-Cr-Fe system has almost identically decorated vertices, which is exactly the case of traditional unit cells. Hence, our findings further unify the concept of units of ordered solids including both quasicrystals and traditionally periodic solids. Therefore, the combination of Gummelt's coverage model and the D_{3H+1BT} reported here enriches the quasi-unit-cells of quasicrystals with perfect and imperfect quasiperiodic

orderings. Consequently, the consistence of quasicrystals and periodic crystals are not only in the sharp diffraction spots in reciprocal space, but also in the single repeating unit in real space. A detailed discussion of the diversities and communalities of these two basic decagonal clusters, the Gummelt and the Lück decagon will be the topic of a forthcoming paper by Steurer.

Minor concerns

1) Methods section is extremely short and lacks in detail. For instance, why was the $\text{Al}_{72}\text{Cr}_{16}\text{Fe}_{12}$ composition chosen? What was the induction furnace model and manufacturer? What temperature was used for the melting process? Did the authors check the final chemical composition of the sample? Why two annealing temperatures? Powder samples were used for TEM analysis, but what was the sample preparation procedure? Authors say that EDS was used to check the chemical composition during TEM analysis. What was the chemical composition of the quasicrystal (not the nominal, but real composition)?

Response: Decagonal quasicrystals were previously reported in the $\text{Al}_{72}\text{Cr}_{16}\text{Fe}_{12}$ alloy, but the structure is unknown [Pavlyuchkov, D. et al. Stable decagonal quasicrystals in the Al-Fe-Cr and Al-Fe-Mn alloy systems. *J. Alloys Compd.* **477**, L41-L44 (2009)]. So we chose the composition of $\text{Al}_{72}\text{Cr}_{16}\text{Fe}_{12}$ for a careful study of crystal structures of DQC. The ZG-001 induction furnace of Liaoning Jinzhou Electric Furnace Co., Ltd. was used for melting high-purity elements at 1500°C , and the SX-G04133 electric box furnace of Tianjin Zhonghuan Furnace Corp was used for heat treating. In order to get the DQC with high quality, we did a series of heat treatment experiments. At last we found that through two heating processes: firstly, annealing at 1025°C for 7 days, and secondly annealing at 1000°C for 7 days, followed by quenching in the water, the quality of DQC would be higher. Therefore, we adopted two annealing temperatures. The composition of DQC was identified as $\text{Al}_{74}\text{Cr}_{15}\text{Fe}_{11}$ by transmission electron microscopy. Powder samples were adopted for TEM observations. We firstly crushed small blocks from the ingot into powders. Then, alcohol was added into the powders to prepare suspension by following ultrasonic for 3 min. Finally, a drop of suspension was dripped onto a 3 mm copper grid covered by hollow carbon film for TEM observations. We added the above information on P3 and P10 in the revised version.

2) In figure 1c authors claimed the existence of a c-glide due to extinctions and the

authors defined the 5D symmetry as $P10_5/mmc$. Authors should detail the diffraction features that led to the conclusion of a presence of the screw-axis as well.

Response: The odd diffraction spots along the tenfold direction in Fig. 1c are extinct, implying a 10_5 screw axis along this direction.

REVIEWERS' COMMENTS

Reviewer #1 (Remarks to the Author):

The revised manuscript has been greatly improved. However, some points in the previous review have not been completely answered. The revised manuscript may be published but it can be achieved to higher quality if the authors consider the following points.

Given at least a structural model which may force the tilting given in the manuscript. Similar to the original Penrose tiling, the matching rules on tiles will force the formation of the Penrose tiling. Overlapping should not be considered as a matching rule.

No discussion on the local structure as matching rules will weak the paper as a novel kind of decagonal ordering.

Reviewer #2 (Remarks to the Author):

The authors have answered the questions and made the respective corrections in their revised manuscript. In the reviewer's opinion, the paper can be accepted for publication.

REVIEWERS' COMMENTS

Reviewer #1 (Remarks to the Author):

The revised manuscript has been greatly improved. However, some points in the previous review have not been completely answered. The revised manuscript may be published but it can be achieved to higher quality if the authors consider the following points.

Given at least a structural model which may force the tilting given in the manuscript. Similar to the original Penrose tiling, the matching rules on tiles will force the formation of the Penrose tiling. Overlapping should not be considered as a matching rule. No discussion on the local structure as matching rules will weak the paper as a novel kind of decagonal ordering.

Response: Thank you for your valuable and thoughtful comments.

Tiling models are no more used for the explanation of the formation and growth of real quasicrystals. Chemically and physically more realistic models are based on the packing of clusters. And the 'matching rules' for quasiperiodic cluster packings are their 'overlapping rules'. They are used for clusters based on Gummelt decagons as well as for the Lück decagons discussed in the present manuscript. For a realistic growth model see, e.g., Kuczera, P. & Steurer, W. (2015). *Phys. Rev. Lett.* **115**, 085502. We added some more discussion at this point in the manuscript.

Reviewer #2 (Remarks to the Author):

The authors have answered the questions and made the respective corrections in their revised manuscript. In the reviewer's opinion, the paper can be accepted for publication.

Response: Thanks again for the reviewer's comments and time to this paper.